# Chemical, Cytotoxic, and Anti-Inflammatory Assessment of Honey Bee Venom from *Apis mellifera intermissa*

**DOI:** 10.3390/antibiotics10121514

**Published:** 2021-12-10

**Authors:** Iouraouine El Mehdi, Soraia I. Falcão, Mustapha Harandou, Saïd Boujraf, Ricardo C. Calhelha, Isabel C. F. R. Ferreira, Ofélia Anjos, Maria G. Campos, Miguel Vilas-Boas

**Affiliations:** 1Faculty of Medicine and Pharmacy, University Sidi Mohamed Ben Abdellah, Fez 30000, Morocco; iouraouine@gmail.com (I.E.M.); harandoumustapha@yahoo.fr (M.H.); sboujraf@gmail.com (S.B.); 2Centro de Investigação de Montanha (CIMO), Instituto Politécnico de Bragança, Campus de Santa Apolónia, 5300-253 Bragança, Portugal; sfalcao@ipb.pt (S.I.F.); calhelha@ipb.pt (R.C.C.); iferreira@ipb.pt (I.C.F.R.F.); 3Observatório de Interações Planta-Medicamento, Faculdade de Farmácia, Universidade de Coimbra, Azinhaga de Santa Comba, 3000-548 Coimbra, Portugal; mgcampos@ff.uc.pt; 4Instituto Politécnico de Castelo Branco, Quinta da Senhora de Mércules, 6001-909 Castelo Branco, Portugal; ofelia@ipcb.pt; 5Centro de Estudos Florestais, Instituto Superior de Agronomia, Universidade de Lisboa, Tapada da Ajuda, 1349-017 Lisboa, Portugal; 6Centro de Biotecnologia de Plantas da Beira Interior, Quinta da Senhora de Mércules, 6001-909 Castelo Branco, Portugal; 7Coimbra Chemistry Centre (CQC, FCT Unit 313) (FCTUC), University of Coimbra, Rua Larga, 3004-535 Coimbra, Portugal

**Keywords:** bee venom, *Apis mellifera intermissa*, anti-inflammatory activity, cytotoxic activity, chemical composition, NIR

## Abstract

The venom from *Apis mellifera intermissa*, the main honey bee prevailing in Morocco, has been scarcely studied, despite its known potential for pharmacological applications. In the present work, we investigated the composition, the anti-inflammatory activity, and the venom’s cytotoxic properties from fifteen honey bee venom (HBV) samples collected in three regions: northeast, central, and southern Morocco. The chemical assessment of honey bee venom was performed using LC-DAD/ESI/MS^n^, NIR spectroscopy and AAS spectroscopy. The antiproliferative effect was evaluated using human tumor cell lines, including breast adenocarcinoma, non-small cell lung carcinoma, cervical carcinoma, hepatocellular carcinoma, and malignant melanoma. Likewise, we assessed the anti-inflammatory activity using the murine macrophage cell line. The study provides information on the honey bee venom subspecies’ main components, such as melittin, apamin, and phospholipase A2, with compositional variation depending on the region of collection. Contents of toxic elements such as cadmium, chromium, and plumb were detected at a concentration below 5 ppm, which can be regarded as safe for pharmaceutical use. The data presented contribute to the first study in HBV from *Apis mellifera intermissa* and highlight the remarkable antiproliferative and anti-inflammatory effects of HBV, suggesting it to be a candidate natural medicine to explore.

## 1. Introduction

Drug discovery has largely benefitted from the development of venomics as a research field, improving, simultaneously, the knowledge on the envenomation processes. Honey bee venom (HBV) is a unique weapon in the animal kingdom which has a prime role in the defense of the bee colony. It is an efficient substance with a unique biocomplexity mixture that exhibits various pharmacological, toxicological, and pleiotropic properties. At least 18 active components, including enzymes, peptides, and biogenic amines, have been discovered in their composition. The main components of HBV include melittin (the primary compound), apamin, secapin, procamine, adolapin, and mast cell degranulating peptide (MCDP). Phospholipase A2 (PLA2) is the primary HBV enzyme, while hyaluronidase, acid phosphomonoesterase, lysophospholipase, and α-glucosidase are also present, but in a much lower amounts. Other non-peptide components affecting many cellular systems can be detected in HBV, including histamine, dopamine, norepinephrine, sugars such as fructose and glucose, and phospholipids [1,2,3,4,5,6,7,8,9,10,11]. HBV is under investigation for different diseases as a fundamental or adjuvant treatment, such as Parkinson’s disease, amyotrophic lateral sclerosis, multiple sclerosis, hepatic fibrosis, atherosclerosis, acne vulgaris, pain, lupus nephritis, periodontal diseases, and several cancers. HBV acts on many inflammatory pathways that can affect the body’s immune system and enhance the differentiation of human regulatory T cells, which play an essential role in controlling SARS-CoV infections [2,12,13,14,15,16,17,18]. The presence of high levels of melittin also potentiates the use of HBV as a fungicide, with a broad inhibition spectrum of the fungal genera, including Aspergillus, Botrytis, Candida, Colletotrichum, Fusarium, Malassezia, Neurospora, Penicillium, Saccharomyces, Trichoderma, Trichophyton, and Trichosporonains [19]. The effect of this biopeptide to bacteria, individually or in consortium with other antibiotics, is also significant, particularly in Gram-positive strains, through the weakening of the cell membrane [11,20].

*Apis mellifera intermissa*, “called the Tellian,” is the endemic honey bee subspecies in North Africa (from Libya to Morocco). Its name comes from its position between tropical African and European species. It is the main abundant honey bee in Morocco where others, such as *Apis mellifera sahariensis* and *Apis mellifera major*, can also be found. *Apis mellifera intermissa*, depicted in Figure 1a, is distributed between the Atlas and the Mediterranean or Atlantic Coast. It is a strong bee with a 6.4 mm tong length, long body, dark pigment, and sparse hairs [21,22,23]. Despite the high potential of bee venom as a natural medicine and the quality being deeply dependent on the honey bee species [24], there are no studies on the compositional variability and bioactivity of bee venom from this honey bee subspecies.

In our investigation, fifteen HBV samples were collected from *Apis mellifera intermissa* apiaries in three distinct regions, namely, northeast, central, and southern Morocco, depicted in Figure 1b, chemically characterized using LC-DAD/ESI/MSn and evaluated for their cytotoxic and anti-inflammatory properties. Furthermore, all samples were submitted to metal determination and explored for their near-infrared profiles as two unmissable tests for standardization and authentication of the bee subspecies’ venom.

## 2. Results and Discussion

### 2.1. Peptide and Enzymatic Composition of Honey Bee Venom

In this work, LC-ESI/MS^n^, in the positive ion mode, was selected due to its simplicity for characterizing the main components of HBV, including apamin, PLA2 and melittin [25]. Although an LC-MS analysis of HBV peptides is complex due to its high molecular weight and multiple charges, the mass spectrum for each chromatographic peak was accessible in the positive ion mode. Good sensibility was achieved since peptides such as melittin and apamin, due to their alkalinity and polarity, easily form protonated molecules in a low pH solution, which can be analyzed in this ion mode. The MS information obtained was in accordance with the previously reported data [14,25]. The chromatograph with the full MS spectra for apamin (MW = 2032 Da) and melittin (MW = 2846 Da) can be observed in Figure 2. For apamin, the double, triple, quadruple and quintuple charged molecular ions with *m*/*z* 1014 ([M + 2H]^2+^), *m*/*z* 677 ([M + 3H]^3+^), *m*/*z* 508 ([M + 4H]^4+^), and *m*/*z* 407 ([M + 5H]*^5+^*) were observed, while for melittin, double, triple, quadruple, quintuple, and sextuple charged molecular ions where found, with *m*/*z* 1424 ([M + 2H]^2+^), *m*/*z* 950 ([M + 3H]^3+^), *m*/*z* 713 ([M + 4H]^4+^), *m*/*z* 571 ([M + 5H]^5+^), and *m*/*z* 476 ([M + 6H]^6+^) [14,25]. The full MS of PLA2 (18.964 Da) is not shown due to the complexity of the obtained data.

The chromatographic profile of the samples from the different regions displayed similar qualitative patterns, represented in Figure 2a, with peaks at 4.7 min, 8.5 min, and 9.7 for apamin (1), PLA2 (2), and melittin (3), respectively. Differences were only observed with respect to quantification, (Table 1). Analogous results obtained in prior studies were added to the table for comparison. HBV from *Apis mellifera intermissa* northeastern samples showed similar values to those of *Apis mellifera iberiensis* [26], but the samples from the other two regions revealed lower averages. 

Conversely, the amount of melittin is the highest among all the studies, which enhances the potential of this HBV to fight against fungal infections [19]. Indeed, melittin is known to interact with several fungus, in a concentration/time manner, through distinct mechanisms that include apoptosis induction by reactive oxygen species accumulation, the inhibition of (1,3)-β-D-glucan synthase, changes in fungal gene expression or by inducing changes in the permeability of the cell wall. This latter mechanism is the same used to justify the high activity of this biopeptide against bacteria, particularly Gram-positive strains: the absence of lipopolysaccharides in the cell wall of Gram-positive bacteria allows for the penetration of melittin within the cell membrane, causing severe disruption on the phospholipids’ packing, thus leading to cell lysis [11,20].

As claimed in previous reports, the concentration of peptides and enzymes in HBV is influenced by the honey bee specie, climate, and geographic conditions [24]. Data analyses between regions, presented in Appendix A, showed a homogeneity of variances for apamin and PLA2, but this feature is violated for melittin. Overall, no significant differences were observed between regions for apamin and melittin. Two samples from the southern region, S3, and S5, recorded the highest value and the lowest values in terms of apamin and melittin (1.45 ± 0.02; 2.55 ± 0.03) and (62.34 ± 0.84; 87.70 ± 0.58), respectively. For PLA2, significant differences were found between northeastern and southern regions. The highest value was recorded in the NE1 sample (9.13 ± 0.13 µg/mL) and the lowest value in the S2 sample (4.18 ± 0.03 µg/mL).

### 2.2. Metal Content of Honey Bee Venom

Unfortunately, information on the metal composition of HBV available in the literature is scarce and is mainly related to the toxic elements classified by the European Medicines Agency (EMA) and the United States Pharmacopeia (USP), including lead (Pb), mercury (Hg), bismuth (Bi), arsenic (As), antimony (Sb), tin (Sn), cadmium (Cd), silver (Ag), copper (Cu), molybdenum (Mo), vanadium (V), Palladium (Pd), platinum (Pt), gold (Au), and ruthenium (Ru) [30,31]. To establish a quality control procedure for HBV as a raw material for pharmaceutical use, it is crucial to identify minerals and to assess which elemental impurities need to be submitted to quality control at a pharmaceutical level. We adopted, in this framework, three essential documents, including the EMA/ICH Q3D guideline for elemental impurities, the USA Ph (231), and Ph. Eur. (2.4.8) chapter on the determination of heavy metals [32,33]. 

Table 2 presents the spectroscopic analyses of HBV samples collected from the three regions under study, i.e., northeastern, central, and southern Morocco. Overall, potassium was found to be the major microelement present in our HBV samples, with a concentration of around 2600 ppm, followed by calcium, sodium, zinc, and magnesium. All these elements (with low inherent toxicity), in addition to lead (human toxicant), presented significant differences between regions (Appendix A). 

The samples from the southern region showed the lowest average content for all microelements, while those from central and northeast Morocco revealed similar levels. It is interesting to note that the level found in samples within each region is remarkably similar, which may reflect the external characteristics of the area. Other metals subjected to our investigation were not always present in all samples from the different sites under study, however, some observations should be described. Generally, the mineral content is in the range of those reported in the literature; however, sodium and calcium levels were 2 to 3 folds higher. Micronutrients or trace elements (toxic at increased concentrations) of Mn, Cu, and Ni were also identified in the samples but in relatively low amounts. The northeast and center samples presented the highest mean values for almost all elemental impurities: heavy metals (Pb and Cr) were higher in the samples from the northeast region but similar to some extent to those found by Kokot et al., 2008 [30]. The presence of elemental impurities may be a result of two categories of sources, i.e., those arising from the environment, including soil nature, nearby mines, fertilizer and pesticides use, or those related to the HBV-collecting process, including combustible material nature used for smoke generation, glace plates and scarper nature, closure and the container system used for product conservation [30]. In this case, the higher level of plumb may be driven by environmental contamination, since the presence of coal mines is common in the northern region of Morocco.

Even though some samples showed significant concentrations of heavy metals that are mentioned in the ICH Q3D and the US Pharmacopeia [34], their levels are still within the recommended ranges (≤5 ppm), indicated by the reference materials noted above in Table 3.

### 2.3. NIR Evaluation of Honey Bee Venom

NIR spectroscopy is a common analytical technique used to identify and quantify the chemical composition of different products in foods, pharmaceuticals, the petroleum industry, and other fields. Its versatility, non-destructive, non-invasive, fast, and precise method made it a useful analytical technique [35,36,37]. Those particularities and its ability to simultaneously evaluate several parameters, led us to explore its potential in the quality control of HBV. 

According to the NIR spectra, there is noticeable information in the region from 9000 to 3500 cm^−1^, Figure 3. The absorption bands for polypeptides (the main compounds present on dry HBV) appear roughly between 4500 cm^−1^ and 6800 cm^−1^ in the overtone region [38,39]. The band at 6591 cm^−1^ corresponds to CH_2_ and CH_3_ linking and to the first overtone of N-H linking [38,39]. Bands between 5749 cm^−1^ and 5138 cm^−1^ represented the combination of O-H stretch and O-H deformation and bent the second overtone [35,38,39]. The band at 4862 cm^−1^ is assigned to the vibration mode of the N-H of the amide II and amine [38,39]. 

For further evaluation, the PCA analysis and second derivative were applied to the more relevant spectral region, according to a previously adopted procedure [35]. Figure 4A1,A2 refer to the entire set of samples, while the results in Figure 4B1,B2, were obtained excluding sample NE5 (sample collected one year before the others). All the HBV samples exhibit a similar spectrum, as shown in Figure 4A2, nonetheless, the ability of this methodology to discriminate among old and fresh bee venom samples is illustrated in Figure 4A1. The region most relevant for this discrimination is between 5000 and 4800 cm^−1^, associated with the combination of O-H stretch and O-H deformation and stretching of C=O bonded to the NH of the peptide linkage, and should correspond to the bonds most susceptible to degradation. Due to the aging difference between this and the remaining samples, the PCA analysis was applied for the fresh samples only, (Figure 4B1,B2). The new PCA score plot (Figure 4B1), can distinguish three groups (a group for each region), while excluding samples NE4, C4, and S4. This exclusion can be explained by the significant differences in PLA2 composition between the three regions, (Appendix A). 

### 2.4. Biological Activity

#### 2.4.1. Cytotoxic Activity

Cell-based cytotoxic activity occurs in an earlier stage to improve the efficacy and safety of new chemical entities in drug discovery [40,41]. For in vitro cell culture systems, a compound or treatment is considered cytotoxic if it interferes with cellular attachment, significantly alters morphology, adversely affects cell growth rate, or causes cell death [42]. Because no assay technology for detecting cytotoxicity in vitro is perfect [42], two conventional assays are usually applied considering their easy operation and standardized readout, namely, electrochemical methods that record impedance related to the physiological status of incubated cells on the gold microchips, and optical methods that measure the absorbance of cell viability-sensitive dyes in the tested solution [43].

HBV has been found to manifest anticancer activity on a range of animal and human tumour cell lines, including mouse melanoma K1735M2 [44], human hepatoma cell line SMMC-7721 [45], human cell leukemic U937 [46], human melanoma A2058 cells [47], malignant melanoma B16F10 cells [48], non-small lung cancer cells A459 [49], and human ovarian cancer cell A2780cp [50]. For this study, we chose the sulforhodamine B (SRB) assay, a conventional colorimetric assay (optical method), applied in the National Cancer Institute (NCI) in the USA and one that integrates the NCI’s compound screening program [51,52]. The cytotoxic activity of HBV was evaluated on five human tumour cell lines, including MCF-7, NCI-H460, HeLa, HepG2, and MM127. Additionally, PLP2 was used to analyze the hepatotoxicity study, since mammalian hepatocytes still represent an obligatory step in evaluating toxic compounds that lead to the production of various metabolites, which are the ultimate cause of toxicity [53]. The obtained results for the five tested tumour cell lines are reported as IG_50_ (µg/mL) in Table 4. 

All samples, independent of the region, demonstrated cytotoxic activity for almost all the studied cell lines (Figure 5). Significant differences have been observed between regions for MCF-7 and MM127 and similarities to some extent have been identified for HepG2, NCI-H460, and HeLa (Appendix A). It seems that HBV samples from the northeastern region had the most potent cytotoxic activity on the different cell lines with the lowest GI_50_. Sample NE4 corroborated this trend well with the smallest GI_50_ values on various cancer cell lines (two to five folds more significant). Sample C3 is an exception, with the best performance on cytotoxicity against HepG2 cell line with a GI_50_ = 1.9 µg/mL, even lower than that obtained with bee venom from north Portugal with a GI_50_ ranging from 5.43 µg/mL to 12.19 µg/mL [26]. Those samples revealed a high combination of melittin and PLA2, which may justify its behavior. In comparison to previous studies, the venom from this specie proved to possess an exceptional cytotoxic activity on NCI-H460, HepG2 but had comparable GI_50_ values for HeLa and MCF-7 [14,26,49,54,55]. It is, however, important to highlight that a significant toxicity in PLP2 is also observed, and so, the potential application of HBV for pharmaceutical proposals must be balanced between the cytotoxic effect in the tumour cells and the lowest toxicity for PLP2 cells. So, if we review the results of Figure 5, the performance of sample NE3, although with half the effectiveness against tumour cells compared to NE4 (double GI), shows seven times more selectivity for tumour cells than PLP2, making it a better candidate for future applications.

#### 2.4.2. Anti-Inflammatory Activity

Anti-inflammatory properties of HBV have been well-proven through in vivo and in vitro assays [10,16,26,56,57,58]. In this study, we evaluated the HBV anti-inflammatory potential of *A. mellifera intermissa* by assessing this substrate’s effects on the pro-inflammatory response in the RAW264.7 macrophage cell line. The results are reported in Table 4, and described as IC_50_ (µg/mL). The anti-inflammatory activity of all the samples is evident, but with a significant difference between the performance of samples from southern and northeastern regions (*p* ≤ 0.05), (Appendix A). The best inhibitory activity on NO production was demonstrated by samples from the northeast region, followed by samples from the center. In contrast, those from the southern region held the lowest activity (Table 4). It is evident that sample NE4 showed a lower IC_50_ value, better than that obtained for *Apis mellifera iberiensis*, 6.2 µg/mL [26], but the result is even more significant when compared with the result obtained for the standard dexamethasone 15.5 µg/mL. Additionally, we noted that the anti-inflammatory activity of sample NE5 was not affected by time, which is a good indicator of its conservation. Conversely, samples S4 and S5 showed the weakest activity with an IC_50_ almost three times higher than the general performance of all the samples, which may be associated with the lowest content of melittin and PLA2. Besides, this is consistent with the known anti-inflammatory activity of apamin and partially of melittin, in addition to synergistic and complementary cytotoxic properties for melittin and PLA2 [4,59,60,61,62]. This behavior becomes evident with the Pearson’s correlation results, representing in Table 5, showing a negative relationship between RAW264.7 and PLA2 and a moderately negative correlation with apamin.

## 3. Materials and Methods

### 3.1. Sample Collection

Fifteen HBV samples were collected from three regions of Morocco, five samples from each region between August and November 2018, Figure 1b). The specific coordinates of the apiaries can be found in the Appendix A. Sample NE5 was collected in the same region, but in 2017, and was used to evaluate storage impact. All samples were collected using a double-face bee venom collector, developed in our laboratory with some specific features, and subjected to a patent. The device was placed in the hive at one of the outmost, opposite ends of the beehive. A mild electrical impulse shock was applied (increasing from 0 V to 12 V, then decreases to initial voltage). The beehive’s optimum duration for bee venom collection was set between 30 min to 60 min, early in the morning or at the beginning of the sunset. The venom collection event’s optimum interval in the same colonies was set between 10 to 15 days. After the collection session, the venom was scraped off from the glass with a sharp scraper and conditioned in pharmaceutical-grade vials. The bee venom was then freeze-dried in a Labconco FreeZone 4.5, Labconco corporation (Kansas City, MO, USA), and kept at −20 °C until further analysis.

### 3.2. Standards and Reagents

Apamin (purity 98.3%) was obtained from CalBiochem (San Diego, CA, USA), while melittin (purity ≥ 85%, HPLC), PLA2 from bee venom (activity: 1775 units mg^−1^ solid), cytochrome *c* from the equine heart (purity ≥ 95%), lipopolysaccharide (LPS), ellipticine, dexamethasone (DM), sulforhodamine B, trypan blue, trichloroacetic acid (TCA), and tris were purchased from Sigma Chemicals Co. (St. Louis, MO, USA). Formic acid was from Panreac (Barcelona, Spain). Standards for metal analysis were purchased from PanReac (Barcelona, Spain). The Griess Reagent System Kit was purchased from Promega (Madison, WI, USA). Dulbecco’s Modified Eagle’s medium (DMEM), Hank’s balanced salt solution (HBSS), Foetal bovine serum (FBS), L-glutamine, trypsin-EDTA, penicillin/streptomycin solution (100 U/mL and 100 mg/mL, respectively) were purchased from TermoFisher Scientific (Waltham, MA, USA). Water was treated in a Milli-Q water purification system (TGI Pure Water Systems, Sunnyvale, TX, USA).

### 3.3. Cell Lines

Both the MM127 and RAW264.7 cells were purchased from the European Collection of Authenticated Cell Cultures (ECACC) (Salisbury, UK), while MCF-7, HeLa, HepG2 and NCI-H460 cells were provided by DSMZ (Leibniz-Institute DSMZ-Deutsche Sammlung von Mikroorganismen und Zellkulturen GmbH). The PLP2 (porcine liver primary culture) was prepared from a freshly harvested porcine liver, obtained from a local slaughterhouse, according to an established protocol described by Abreu et al. [53].

### 3.4. Chemical Characterization of the Samples by LC-ESI/MS^n^

The LC-DAD-ESI/MS^n^ analyses were performed on a Dionex Ultimate 3000 UPLC instrument (Thermo Scientific, San Jose, CA, USA), equipped with a diode-array detector and coupled to a mass detector. The chromatographic system consisted of a quaternary pump, an autosampler maintained at 5 °C, a degasser, a photodiode-array detector, and an automatic thermostatic column compartment. The chromatographic separation was conducted according to a previously described method [26], using an XSelect CSH130 C18, 100 mm × 2.1 mm id, 2,5 µm XP column (Waters, Milford, MA, USA), with the temperature maintained at 30 °C. Spectral data of all peaks were accumulated in the range of 190–500 nm. Cytochrome *c* was used as an internal standard (IS) and prepared in deionized water at the concentration of 25 µg/mL. The lyophilized HBV (3 mg) was dissolved in 10 mL of IS solution for the analysis. Each sample was filtered through a 0.2 µm nylon membrane (Whatman). The standard solutions were prepared by dissolving the compound in an IS solution at the desired concentration. Quantification was achieve using calibration curves for apamin (4–60 µL/mL; y = 0.031x + 0.021; *R*^2^ = 0.998), PLA2 (8–120 µL/mL; y = 0.026x + 0.118; *R*^2^ = 0.996) and melittin (31–500 µL/mL; y = 0.034x + 0.026; *R*^2^ = 0.999). The mass spectrometer was operated in the positive ion mode using Linear Ion Trap LTQ XL mass spectrometer (Thermo Scientific, San Jose, CA, USA) equipped with an ESI source. Typical ESI conditions were nitrogen sheath gas 35 psi, spray voltage 3.5 kV, source temperature 300 °C, capillary voltage 20 V, and the tube lens offset was kept at a voltage of 74 V. The collision energy used was 30 (arbitrary units). Data acquisition was carried out with the Xcalibur^®^ data system (Thermo Scientific, San Jose, CA, USA).

### 3.5. Metal Content by Atomic Absorption Spectroscopy (AAS)

The analysis of minerals and heavy metals was performed using a Perkin Elmer PinAAcle 900T (PerkinElmer Inc., MA, USA) atomic absorption spectrometer. For potassium (K), sodium (Na), calcium (Ca), magnesium (Mg), copper (Cu), and zinc (Zn), the quantification was performed with flame atomic absorption spectroscopy, while for the determination of cadmium (Cd), manganese (Mn), nickel (Ni), and lead (Pb) the analysis was performed on a graphite chamber using Zeeman for background correction.

The standards used for the calibration were prepared daily by dilution from stock solutions (1000 ppm). For microwave digestion, 1 mg of HBV was weighed into a PTFE digestion tube and added to 10 mL of HNO_3_. The digestion was performed in a CEM Mars 5 microwave (CEM co, Matthews, NC, USA), using the following ramp temperature program: for 15 min until 200 °C with a power of 1200 W. These conditions were maintained for an additional 15 min. After cooling, the digested samples were diluted to a final volume of 50 mL with deionized water to analyze the minerals and heavy metal content.

### 3.6. Near-Infrared Analysis of Honey Bee Venom (NIR):

The NIR spectra of the samples were obtained according to the methodology proposed previously by Yang et al. [63], using an NIR spectrometer (MPA Bruker Optik, Germany) operated with the OPUS^®^, version: 7.5.18 software in reflectance mode. Approximately 10 mg of HBV was poured into a clean glass vial with a 22 mm diameter. The samples were measured with a spectral resolution of 8 cm^−1^ and 32 scans, in the wavenumber range between 3600 and 10,000 cm^−1^.

### 3.7. Citotoxic Activity

The HBV samples were dissolved in water at 4 mg/mL and then submitted to further dilutions from 4 to 0.0625 mg/mL. The cytotoxic effects were evaluated using five human cell lines, namely, MCF-7 (breast adenocarcinoma), NCI-H460 (non-small cell lung cancer), HeLa (cervical carcinoma), HepG2 (Hepatocellular carcinoma), and MM127 (human malignant melanoma) from the ECACC General Cell Collection. Cells were routinely maintained as adherent cell cultures in RPMI-1640 medium containing 10% heat-inactivated FBS and 2 mM glutamine, 100 U/mL penicillin, and 100 µg/mL streptomycin, and kept at 37 °C in a humidified air incubator containing 5% CO_2_. According to the procedure described previously, the cytotoxic potential of the HBV was evaluated by the sulforhodamine B assay [64,65]. For the hepatotoxicity evaluation, a cell culture was prepared from a freshly harvested porcine liver, obtained from a local slaughterhouse, according to an established procedure, and labeled as PLP2 (porcine liver primary culture) [53]. The cell cultivation was maintained with direct monitoring every two to three days, using a phase-contrast microscope. Before confluence was reached, cells were sub-cultured and plated in 96-well plates at a density of 1.0 × 10^4^ cells/well, cultivated in DMEM medium with 10% FBS, 100 U/mL penicillin, and 100 µg/mL streptomycin. Ellipticine was used as a positive control, and the results were expressed in GI_50_ values (concentration that inhibited 50% of the net cell growth). 

### 3.8. Anti-Inflamatory Activity

The anti-inflammatory potential was evaluated using a murine mouse macrophages (RAW 264.7) cell line [26]. Cells were routinely maintained as adherent cell cultures in DMEM medium containing 10% FBS, 2 mM glutamine, 100 U/mL penicillin, and 100 µg/mL streptomycin and kept at 37 °C in a humidified air incubator containing 5% CO_2_. The HBV samples were dissolved in water at 4 mg/mL and then submitted to further dilutions from 4 to 0.0625 mg/mL. The RAW264.7 cells (1 × 10^5^ cells/mL) were pre incubated for one hour with various HBV concentrations and stimulated with LPS (1 µg/mL) at 37 °C for 18 h in the medium. Nitric oxide (NO) levels were determined by measuring nitrite levels in the culture media using Griess Reagent assay [26]. The results were expressed in percentages of NO production inhibition, compared to the negative control (without LPS), and IC_50_ values equal to the sample concentration providing a 50% inhibition of NO production, were also estimated. Dexamethasone was used as the positive control. Cytotoxicity assays were previously performed to ensure that HBV did not kill cells (results not shown), and so guaranteeing that the decrease in NO production occurs due to the anti-inflammatory effect of the sample under study and not due to the death of RAW264.7 cells.

### 3.9. Data Analysis

All experiments were performed in triplicate and the results are expressed as the mean ± SD. One-way analysis of variance (ANOVA) and Tukey’s HSD Post Hoc test were performed to search for significant differences between regions. Afterward, the Welch test was applied as a robust test of equality of means. A Pearson’s correlation analysis was explored to establish possible correlation links between venom chemical composition and its biological activities, and to assess whether the regional factor produces any affects. Statistical analyses were conducted using computer-based statistical software (IBM, SPSS v. 20.0); a *p*-value ≤ of 0.05 was considered as statistically significant. A principal component analysis (PCA) was applied for the NIR spectra of bee venom. Four replicated spectra, for each sample, were collected and the analysis was performed in the entire region between 9000 and 3600 cm^−1^. The first derivative of Savitzky–Golay was applied as a pre-process [35]. All spectral data analyses were performed using the Unscrambler^®^ X version: 10.5.46461.632 (CAMO Software AS, Oslo, Norway).

## 4. Conclusions

This work presented, for the first time, the characterization of *Apis mellifera intermissa* venom collected in different regions (northeast, center, and southern) of Morocco. Within the most abundant HBV compounds, melittin, PLA2, and apamin, only the enzyme PLA2 was observed at significant differences between regions. This pattern is clearly supported by the data provided by NIR spectroscopy. Particularly relevant is the higher content of melittin in *Apis mellifera intermissa* venom compared to other subspecies, indicative that the honey bee species is more significant for HBV peptide/enzyme composition than the geographical location. The heavy metals and micronutrients or trace elements were found in all samples, but remained in the range recommended by the guideline references for pharmaceutical raw materials. Nevertheless, the results made evident that special care should be taken when defining the HBV collection site. 

All HBV samples displayed cytotoxic and anti-inflammatory activity. Still, samples from the northeastern region, with the highest concentration mean values in apamin and PLA2, manifested significant activities on MCF-7, MM127, and RAW264.7. The findings are consistent, to a large extent, with the known anti-inflammatory activity essential for apamin and partially for melittin, in addition to synergistic and complementary cytotoxic properties for melittin and PLA2. The high melittin content in all regions is also a good indicator of the potential of *Apis mellifera intermissa* venom as a natural antibacterial and antifungal source.

Many predictive in vitro toxicology assays, including cardiotoxicity, central nervous system toxicity, genetic toxicity, and nephrotoxicity, must be established and based on a tiered approach for data collection and interpretation. Some ambient conditions can be pollutant sources, affecting the product quality, including mine closeness, pesticides use, or the conduction of collecting processes that do not respect the good apicultural practices of bee-venom collection. In this context, it was shown that the main components, melittin, PLA2, and apamin, can be used as key elements in the standardization and the authentication of this product. Besides, NIR and AAS can be highly recommended as two reliable techniques for limit tests and the authentication of HBV as a pharmaceutical raw material. 

## Figures and Tables

**Figure 1 antibiotics-10-01514-f001:**
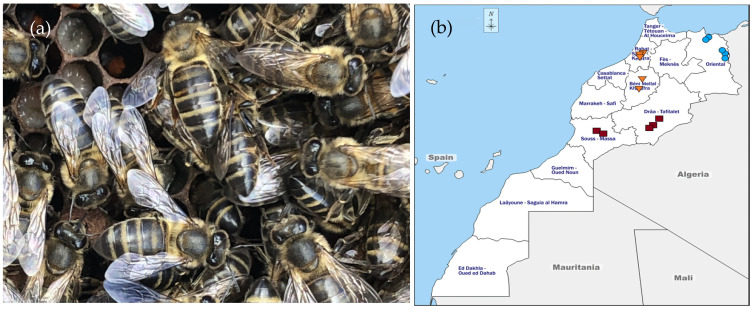
(**a**) *Apis mellifera intermissa*; (**b**) Map of Morocco with the site of collection: 
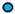
 northeast, 
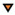
 center and 
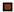
 southern samples.

**Figure 2 antibiotics-10-01514-f002:**
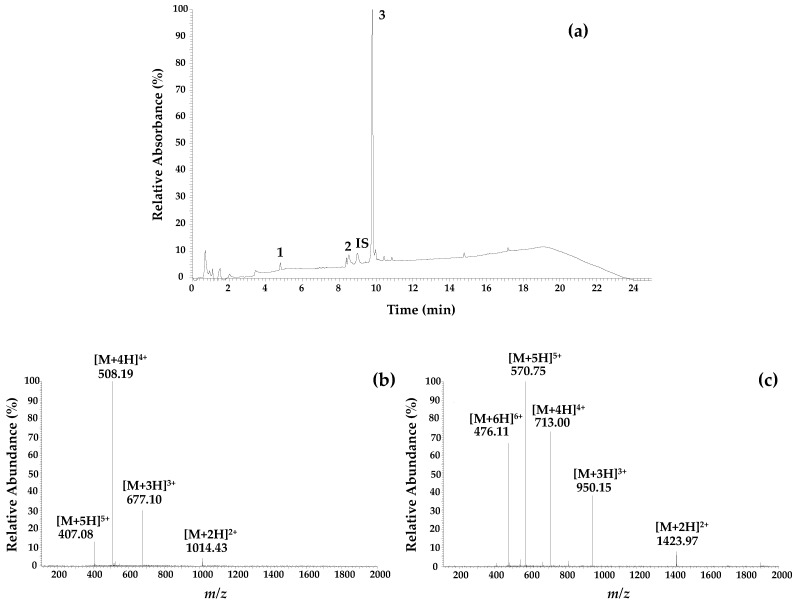
(**a**) Representative chromatographic profile at 220 nm of *A. mellifera intermissa* venom (1-apamin; 2-phospholipase A2; IS-internal standard: cytochrome c at 25 µg/mL; 3-melittin); (**b**) Full scan mass spectrum of apamin (tr = 4.78 min; MW = 2032 Da); (**c**) Full scan mass spectrum of melittin (tr = 9.80 min; MW = 2846 Da).

**Figure 3 antibiotics-10-01514-f003:**
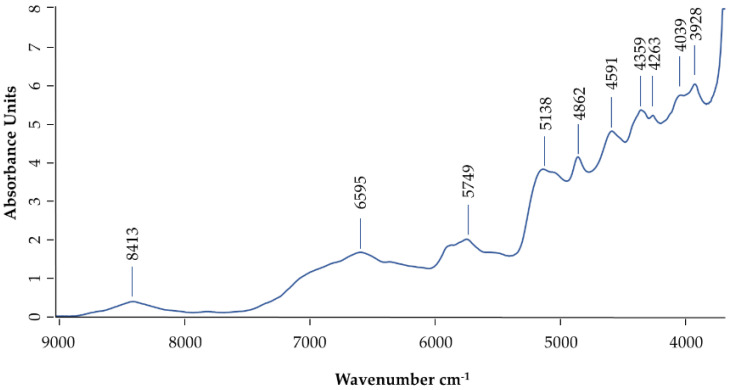
Normalized NIR spectra for HBV, with the indication of the most relevant bands.

**Figure 4 antibiotics-10-01514-f004:**
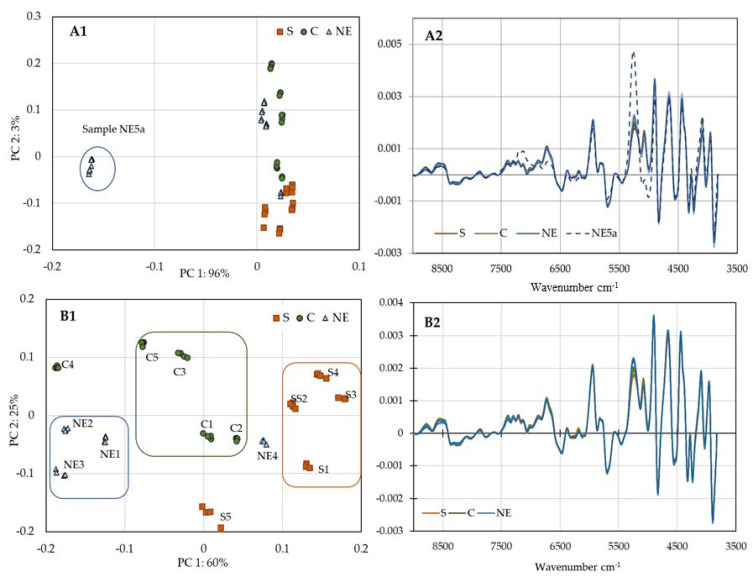
PCA score plot of the more relevant spectral region of NIR spectra of *A. mellifera intermissa* venom from the three regions of Morocco (**A1**) and excluding sample NE5 (**B1**). (**A2**,**B2**) displayed the second derivative pretreatment.

**Figure 5 antibiotics-10-01514-f005:**
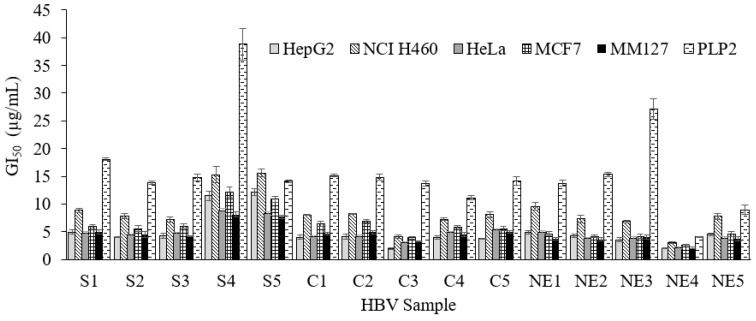
Cytotoxicity activity of *A. mellifera intermissa* venom from the three regions (S, South; C, Center; NE, Northeast Morocco). The results are expressed in GI_50_ values (µg/mL), corresponding to the concentration causing 50% growth inhibition in the five human tumor lines or porcine liver primary culture PLP2.

**Table 1 antibiotics-10-01514-t001:** Chemical characterization of *A. mellifera intermissa* venom from northeast, central, and southern Morocco by LC-DAD-ESI/MS^n^.

Sample	Apamin (µg/mL)	PLA2 * (µg/mL)	Melittin (µg/mL)
NE1	2.08	9.13	67.16
NE2	1.93	6.66	73.43
NE3	2.29	6.57	78.14
NE4	1.99	7.4	73.85
NE5	1.96	8.54	65.97
Average ± SD	2.1 ± 0.1	8 ± 1	72 ± 5
C1	1.86	4.21	71.78
C2	2.05	5.05	73.66
C3	1.96	7.14	71.98
C4	2.10	8.03	75.10
C5	1.99	6.42	70.15
Average ± SD	2.0 ± 0.1	6 ± 1	73 ± 2
S1	1.53	4.52	66.78
S2	1.69	4.18	71.29
S3	2.55	7.81	87.70
S4	1.79	1.79	70.29
S5	1.45	4.45	62.34
Average ± SD	1.8 ± 0.4	5 ± 2	72 ± 9
[26]	1.5	8	65
[27]	4.1	15	58
[28]	2.6	13	54
[29]	3.5	13	65

NE–northeast; C–center; S–Southern Morocco. SD–Standard deviation. * Significant differences between regions for *p* ≤ 0.05.

**Table 2 antibiotics-10-01514-t002:** Metal content of *A. mellifera intermissa* venom from the three regions of Morocco.

Samples	K * (mg/g)	Na * (mg/g)	Ca * (mg/g)	Mg * (mg/g)	Zn * (mg/g)	Cu (µg/g)	Cd (µg/g)	Cr (µg/g)	Mn (µg/g)	Ni (µg/g)	Pb * (µg/g)
NE1	2.97	1.30	3.25	0.52	1.23	5.37	-	5.20	2.02	26.98	6.13
NE2	3.01	1.34	1.78	0.44	1.39	24.72	-	9.46	4.46	4.32	5.15
NE3	2.64	1.53	1.80	0.33	1.05	4.36	-	7.61	1.61	1.36	9.85
NE4	2.16	1.00	0.95	0.32	1.04	-	2.26	3.86	1.10	-	8.08
NE5	3.57	1.55	1.76	0.57	1.03	4.13	-	8.63	2.79	0.49	6.77
Average ± SD	2.9 ± 0.5	1.3 ± 0.2	1.9 ± 0.7	0.4 ± 0.1	1.2 ± 0.2	8 ± 9	-	7 ± 2	2.4 ± 1.2	8 ± 11	7.2 ± 1.6
C1	1.62	1.93	4.55	0.42	1.03	9.45	-	4.59	2.47	2.83	3.81
C2	1.73	1.01	2.16	0.27	1.12	-	-	3.41	1.59	0.64	3.56
C3	3.02	1.84	3.22	0.55	1.42	20.91	-	6.71	3.10	0.62	3.99
C4	3.03	1.43	2.32	0.56	1.27	6.83	-	4.99	2.07	4.49	4.84
C5	2.50	0.98	2.59	0.38	1.53	7.30	-	8.29	3.76	1.78	5.16
Average ± SD	2.4 ± 0.6	1.4 ± 0.4	3.0 ± 0.9	0.4 ± 0.1	1.3 ± 0.2	9 ± 7	-	6 ± 2	2.6 ± 0.8	2.1 ± 1.5	4.3 ± 0.6
S1	1.55	0.79	0.96	0.28	0.93	-	-	2.40	0.78	-	7.60
S2	1.68	0.81	0.97	0.27	1.03	3.95	-	2.45	1.19	-	3.63
S3	1.95	0.81	0.83	0.31	0.95	4.10	1.32	13.21	0.89	-	3.24
S4	1.72	0.92	1.41	0.25	1.07	-	-	5.19	1.29	-	4.32
S5	1.56	0.79	0.95	0.25	1.07	3.95	-	5.55	2.19	-	4.20
Average ± SD	1.7 ± 0.2	0.83 ± 0.05	1.0 ± 0.2	0.3 ± 0.0	1.1 ± 0.1	2.4 ± 2	-	6 ± 4	1.3 ± 0.5	-	4.6 ± 1.6

NE–northeast; C–center; S–southern Morocco. SD–Standard deviation. * Significant differences between regions for *p* ≤ 0.05.

**Table 3 antibiotics-10-01514-t003:** Oral, parenteral and inhalation concentration levels (µgg^−1^), for elemental impurities in the drug substances and excipients set by the ICH guideline Q3D on elemental [34]. Comparison with HBV from the three regions (in mean values).

	Class	Oral	Parenteral	Inhalation	HBV Concentration	
Cd	1	0.5	0.2	0.3	NE	a
					C	-
					S	a
Pb	1	0.5	0.5	0.5	NE	7.3 *
					C	4.3 *
					S	4.6 *
Ni	2A	20	2	0.5	NE	8.2
					C	2.1
					S	-
Cu	3	300	30	3	NE	8.6
					C	8.9
					S	2.4
Cr	3	1100	110	0.3	NE	6.5
					C	5.6
					S	5.8

Class1: Metals of significant safety concerns; Class 2A: Metals with low safety concerns; Class 3: Metals with minimal concerns. NE–northeast; C–center; S–Southern Morocco. a: Detected only in one sample. * Significant differences between regions for *p* ≤ 0.05.

**Table 4 antibiotics-10-01514-t004:** Cytotoxic and anti-inflammatory activity of *A. mellifera intermissa* venom from the three regions of Morocco.

	Cytotoxic Activity (GI_50,_ µg/mL)						Anti-Inflammatory Activity (IC_50_, µg/mL)
	HepG2	NCI_H460	HeLa	MM127 *	MCF7 *	PLP2	RAW264.7 *
NE1	4.86	9.57	4.84	3.72	4.64	13.77	5.07
NE2	4.32	7.35	3.88	3.64	4.2	15.33	4.89
NE3	3.58	6.88	3.75	4.01	4.17	27.15	6.06
NE4	2.40	3.09	2.08	2.05	2.62	4.06	4.03
NE5	4.56	7.77	3.81	3.80	4.55	8.96	4.86
Average ± SD	3.9 ± 0.9	7 ± 2	3.7 ± 0.9	3.4 ± 0.7	4.0 ± 0.7	14 ± 8	5.0 ± 0.7
C1	4.02	8.06	4.15	4.64	6.40	15.13	7.92
C2	4.1	8.26	4.06	4.92	6.86	14.84	7.78
C3	1.99	4.12	2.97	3.33	3.91	13.74	6.12
C4	3.94	7.27	4.94	4.44	5.9	11.13	6.26
C5	3.79	8.08	5.36	4.95	5.59	14.14	6.11
Average ± SD	3.6 ± 0.8	7 ± 2	4.3 ± 0.8	4.5 ± 0.6	5.7 ± 1.0	14 ± 1	6.8 ± 0.8
S1	4.95	8.85	4.67	4.99	5.94	18.03	9.08
S2	3.98	7.85	4.36	4.49	5.52	13.93	7.40
S3	4.33	7.17	4.67	4.24	6.02	14.86	6.65
S4	11.46	15.3	8.71	8.09	12.1	38.85	15.5
S5	15.63	15.6	8.35	7.68	10.8	14.12	15.0
Average ± SD	8.0 ± 4.7	11 ± 4	6.2 ± 1.9	5.9 ± 1.7	8.1 ± 2.8	20 ± 10	10.7 ± 3.8
Dexamethasone							15.5

NE–northeast; C–center; S–Southern Morocco. SD–Standard deviation. GI_50_-concentration that inhibited 50% of the net cell growth. IC_50_-Sample concentration providing 50% of inhibition of NO production. * Significant differences between regions for *p* ≤ 0.05.

**Table 5 antibiotics-10-01514-t005:** Pearson’s correlation between the main HBV component (melittin, PLA2 and apamin) and the cytotoxicity and anti-inflammatory activity GI_50_/IC_50_ (µg/mL).

	HepG2	NCI-H460	HeLa	MCF-7	MM127	PLP2	RAW264.7
Apamin	−0.497	−0.490	−0.409	−0.430	−0.483	−0.053	−0.560 *
PLA2	−0.412	−0.443	−0.391	−0.555 *	−0.590 *	−0.382	−0.652 **
Melittin	−0.311	−0.302	−0.284	−0.214	−0.205	0.154	−0.161

GI_50_-concentration that inhibited 50% of the net cell growth. IC_50_-Sample concentration providing 50% of inhibition of NO production. * Significant correlation at the (0.05) level; ** significant correlation at (0.01 level).

## Data Availability

Data is contained within the article or Appendix A.

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
