# Peer review of "Chemical, Cytotoxic, and Anti-Inflammatory Assessment of Honey Bee Venom from Apis mellifera intermissa"

_antibiotics, 2021, doi:10.3390/antibiotics10121514_

Round 1

Reviewer 1 Report

It is interesting that the composition of some HBV ingredients is various depending on the collected regions.  In addition, the anti-inflammatory activity and the cytotoxicity of HBV on various tumor cells were examined in detail.

However, the data to clear the relationship between the main HBV components (melittin, apamin, and phospholipase A2) and the harmful phenomenon of HBV such as anaphylactic reaction are still insufficient. Therefore, it is difficult to make HBV an excellent candidate to explore as natural antibiotic from the data presented in the study.  The provided data only showed possibility of the use of HBV, and it is too exaggerated to conclude that HBV as an excellent candidate.

Author Response

Please find the answer to all comments in the attached document.

Reviewer 2 Report

This reviewer is satisfied with the content of this study and only has a few comments on this study for authors' information.

1) First of all, the quality (resolution) of the figures is too low and must be improved before acceptance. Some details are barely identified from current figures.

2) It is better to mark the three locations of venom used in this study on a map. Another issue is the origin of the cell lines used. Those are needed information. 

Author Response

Please see the comments in the attached document.

Reviewer 3 Report

Title - Characterisation normally refers to physical and chemical properties. Consider revising the title.

Introduction

The choice of this bee species for the venom is not clearly explained in the Introduction.

Results and Discussion

Authors should start presenting the results. These sentences are not necessary. Perhaps, authors should briefly explain about how many HBV samples were investigated.

Presentation of results and discussion could have been improved by stating the findings first, and followed with the interpretation of the findings. 
Current presentation is difficult for readers to follow. 
This recommendation applies to all subsections of Results and Discussion. 

A table should be self-explanatory. Abbreviations must be described in full below the table.

Line 118 "what potentiates its use to fight against fungal infections" - Needs further explanation.

2.3 NIR evaluation of honey bee venom - This paragraph lacks discussion in relation to the findings.

Materials and Methods

Authors should consider providing clear explanation on how to conduct the experiments. Detail description of the specifications but lack explanation of the methodology itself. 

3.8 Data analysis - Why duplicate and not triplicate? This would have serious repercussion on the validity of the inferential statistics. 

Conclusions - Very lengthy. Repetition of results. Authors should focus on making the conclusions based on the findings and the interpretations.  

General comments: 

There are typo and spelling errors throughout the manuscript that require attention. 

The manuscript would benefit from a thorough proofreading. 

Author Response

(The authors gave the same response as above.)

Reviewer 4 Report

The authors investigated the chemical composition of Apis mellifera intermissa venom (HBV) collected from northeast, central and southern Morocco. The chemical characterization has been performed by complementary analytical techniques and evidenced significant differences in quantitative but not qualitative composition. These differences were then liked to different in vitro activities, namely cytotoxic and anti-inflammatory properties.

The study does not fit with the scope of Antibiotics. The methodology used by the authors is not robust, as the sample is too limited (only five replicates for each region). The experiments have to be improved. Overall, the manuscript is well written, but the presentation of results is not always of quality. I have the following observations.

Major points:

  • Authors report differences in terms of the chemical composition of HBV based on data reported in Table 1. Are these differences significant? If not, it has no sense to discuss what region allows HBV with a higher or lower amount of the considered compound. If yes, the statistical significance has to be highlighted in the table.
  • The authors found significant differences in terms of heavy metals. The statistical significance has to be highlighted in Table 2.
  • The results about higher heavy metals contamination from samples collected in the northeast region should be better discussed. Indeed, only in the conclusion section the authors discuss that they may be due to many coal mines in that region.
  • The authors introduce in section 2.4.1 the importance of cytotoxicity for new chemical entities. While they demonstrate the cytotoxic effect of HBV on tumour cell lines, which is good, they do not investigate the cytocompatibility on non-cancer cells. And that is bad and also compromises the anti-inflammatory results. If HBV has a cytotoxic effect on murine macrophages, a lower amount of nitric oxide is produced once it kills them. But not as the consequence of a lower inflammatory process, but as a consequence of a cytotoxic effect.
  • Furthermore, the statistical significance for cytotoxic and anti-inflammatory results has to be highlighted in Table 4.
  • Figures 5 and 6 are unnecessary as they are unclear and, mainly, they repeat the data already presented in Table 4.

Minor points:

  • Authors should read and revise the manuscript, as part of the Manuscript template is still present in the document’s final version.
  • Please specify the acronym HBV in the abstract.
  • Figure 1 reported is unnecessary. However, if the authors want to highlight the characteristics of Apis mellifera intermissa, a better figure should be provided, where the characteristic length, dark pigment and sparse hair are evident.
  • Please provide more details about the freeze-drying process of HBV.

Author Response

(The authors gave the same response as above.)

Round 2

Reviewer 4 Report

I thank the authors for the changes made to the manuscript according to my comments. However, even after the changes, the study does not fit with the scope of Antibiotics, and thus it should be rejected. Furthermore, the methodology used by the authors is not robust, as the sample is too limited (only five replicates for each region).